# First Serologic Survey of *Erysipelothrix rhusiopathiae* in Wild Boars Hunted for Private Consumption in Portugal

**DOI:** 10.3390/ani13182936

**Published:** 2023-09-16

**Authors:** João Canotilho, Ana Carolina Abrantes, David Risco, Pedro Fernández-Llario, José Aranha, Madalena Vieira-Pinto

**Affiliations:** 1ReproVet, Av. Rainha Dona Amélia, 6300-749 Guarda, Portugal; joaonunocanotilho@gmail.com; 2CECAV-Animal and Veterinary Research Centre, Trás-os-Montes e Alto Douro University, Quinta de Prados, 5000-801 Vila Real, Portugal; carolina.psca@gmail.com; 3Departamento de Medicina Animal, Universidad de Extremadura, 10003 Cáceres, Spain; driscope@gmail.com; 4INGULADOS—Innovación en Gestión y Conservación de Ungulados SL, C. Miguel Servet, 11, 10004 Cáceres, Spain; pfernandezllario@gmail.com; 5CITAB—Centre for the Research and Technology of Agro-Environmental and Biological Sciences, Trás-os-Montes and Alto Douro University, Quinta de Prados, 5000-801 Vila Real, Portugal; j_aranha@utad.pt; 6Department of Veterinary Science, Trás-os-Montes e Alto Douro University, 5000-801 Vila Real, Portugal; 7Associate Laboratory for Animal and Veterinary Sciences (AL4AnimalS), 1300-477 Lisbon, Portugal; 8CISAS—Center for Research and Development in Agrifood Systems and Sustainability, Polytechnic Institute of Viana do Castelo, NUTRIR (Technological Center for AgriFood Sustainability), Monte de Prado, 4960-320 Melgaço, Portugal

**Keywords:** erysipelas, large game, occupational disease, zoonosis

## Abstract

**Simple Summary:**

Swine erysipelas (SE), caused by the bacterium *Erysipelothrix rhusiopathiae*, is a relevant zoonotic disease affecting domestic pigs. However, there are scarce studies on the occurrence of SE in hunted wild boar and, consequently, on its role as a reservoir and spill over to other animals and humans. This lack of knowledge, which also applies to the Portuguese wild boar, triggered the development of this first survey. Of the 111 wild boars sampled, seroprevalence was 16.2% (95% CI: 19.9–24.4%), pointing out the importance of this infection in the wild boar population. Given its zoonotic nature, it can have serious repercussions on people who handle and eviscerate the carcasses of hunted wild boar, especially hunters. Thus, the need for further studies to improve the epidemiological knowledge of ES in the wild boar population in Portugal is emphasized. Moreover, the need to adopt preventive measures and train the various stakeholders, especially hunters, who are in direct contact with these wild populations on a daily basis and are at risk of exposure to this infectious agent, is underlined.

**Abstract:**

*Erysipelothrix rhusiopathiae* is a relevant zoonotic infectious agent causing swine erysipelas (SE) in wild boar. In Portugal, there is no information on its occurrence. For this reason, this study aims to perform a first serosurvey of SE in hunted wild boars in Portugal. During the 2019/2020 hunting season, 111 sera from hunted wild boar were collected and analysed serologically in the laboratory with a commercial ELISA kit. No animals were eviscerated and examined after the hunt. The hunters took it all for private consumption. The results identified 18 animals that were exposed to SE, corresponding to a seroprevalence of 16.2% (95% CI: 19.9–24.4%). No statistical significance was observed on the effect of gender and age on seropositivity. However, wild boar hunted in Pinhel County, had five times more likely to be seropositivity (*p*-value < 0.05; OD = 5.4). Apart from its potential debilitating capacity and chronicity in the wild boar population, SE is also a very serious occupational zoonosis. Thus, the result of this first serosurvey in Portugal should raise awareness and alert competent national veterinary authorities and those involved in the hunting sector, especially hunters who directly handle these carcasses. Further studies should be conducted to better understand the role of wild boar as a reservoir and spillover of this disease to other animals and humans.

## 1. Introduction

The number of Wild boar (*Sus scrofa*) is on the rise in mainland Europe, and which Mediterranean area is not an exception [1]. Currently, in Portugal, it is estimated that there are around 300,000 wild boars throughout the country [2].

For Massei et al. (2015), these population density levels found in Europe result from the fact that their population growth does not appear to be self-limiting and is difficult to control by the current levels of hunting pressure generally performed in Europe. Although they are an important economic resource for many landowners and hunting organisers, as well as many hunters, this growing number of wild boars are being considered as a problem for agriculture, forestry and wildlife conservation [3,4]. Furthermore, being a host of a number of pathogenic agents, wild boar also represents a potential threat of disease transmission [1], some of them zoonotic. Their feeding behaviour, as omnivores and opportunistic scavengers, means that wild boar are considered a species of great importance when analysing the risk of disease dispersal in rural areas, where the domestic-wild interface is marked [5]. A review of viral diseases of the European wild boar can be seen in the manuscript written by Ruiz-Fons, Segalez and Gortazar (2008) and a systematic review of 15 years overview of European zoonotic surveys in wild boar can be found in Abrantes and Vieira-Pinto (2023) [6,7].

In Portugal, wild boars are mainly hunted by drive hunt for about five months (from September to February), and around 30,000 wild boars are hunted per year [8,9]. From those, the majority are for private consumption and are not placed on the market. According to Regulation (EC) No 178/2002, ‘placing on the market’ means the holding of food or feed for the purpose of sale, including offering for sale or any other form of transfer, whether free of charge or not, and the sale, distribution and other forms of transfer themselves [10]. Under this legal framework, private domestic use, it is not included in this definition. This way, by law (Commission Implementing Regulations (EU) 2019/627), all the meat for private consumption does not need to be sent to an approved wild game handling establishment to be subject to an official post-mortem inspection [11], increasing the risk of zoonotic transmission to Human.

One of the zoonotic diseases that may affect wild boar is caused by *Erysipelothrix rhusiopathiae*, a thin, gram-positive, aerobic bacillus belonging to the class *Erysipelotrichia* of the phylum Firmicutes [12].

Although *Erysipelothrix rhusiopathiae*, may affects several domestic and wild animal species [12], suidae family is considered a reservoir of this infection that is called Swine Erysipelas (SE). In domestic pigs, it is referred to as a disease of greatest prevalence and economic importance [13,14].

In the Suidae family, the transmission of *Erysipelothrix rhusiopathiae* may occur directly through ingestion of faeces, urine, saliva and nasal secretions from diseased animals or indirectly through contaminated food or water. Infection can also occur through skin abrasions or through mechanical vectors, such as arthropod bites [13].

Usually, bacteraemia occurs within the first twenty-four hours after contact, and immunosuppressed animals are the most susceptible to contracting the disease [15]. The clinical signs of infection can be divided into three types: acute, subacute, or chronic. The acute phase is characterised by sudden death or signs of generalised infection and septicaemia; sometimes areas with erythema, petechiae, vesicles, and necrosis may appear. They can also have abortions, depression, lethargy, fever and painful joints in this phase. Although in wild boar, the skin lesions can be less detectable because of the boars’ thick hair coat, the lesions found seem not to differ from the acute form of the disease known in domestic pigs [16]. The subacute phase has less severe symptoms than the acute phase, where skin lesions may be absent or with little intensity. The chronic phase usually results from the animal surviving the acute or subacute disease. At this stage, affected animals may have endocarditis and chronic arthritis [13,16], which can be seen if a systematic initial examination is performed by a trained person after hunting. If no death occurs, infected animals with chronic infection may become carriers. Apparently, carriers have this microorganism in the tonsils and other lymphoid tissues of the digestive tract. Under favourable conditions, these bacteria can enter the deeper tissues of the body or the bloodstream. Excretion of *Erysipelothrix rhusiopathiae* may occur through their faeces, urine, saliva and nasal secretions, providing an important source of infection. In this way, soil, bedding, food and water can be contaminated, leading to indirect transmission of the organism [13].

The diagnosis can be based on the clinical signs of the disease. The definitive diagnosis is carried out in the laboratory by isolating the bacteria in samples from different organs, such as the heart, lung, liver, spleen, joints, and kidney. Blood cultures can also be performed during the acute phase. However, in the chronic phase, the diagnosis must be serological. The following tests can perform serological diagnosis: plate, tube and microtiter agglutination; passive hemagglutination; inhibition of hemagglutination; complement fixation; ELISA; and immunofluorescence [13,16,17]. Serologic tests are useful in ascertaining the exposure of animals to SE [13].

It Is well known that wild boar is also susceptible to SE infection [14], but scarce information is known about it and ecology still needs to be studied. To date, no studies on SE prevalence in wild boar in Portugal have been carried out. Information related to its potential influence on the health of domestic pigs is also unknown. However, a recent study in Italy suggests that wild boars could act as SE spillover and there is an association between SE infection in wild species and domestic pigs, linked especially to anthropogenic factors [18].

Like other infectious diseases, such as African Swine fever and Tuberculosis, SE is a disease that circulates at the interface between domestic and wild animals. This risk of intra-species transmission is most marked in outdoor and/or backyard pigs and wild boars [19]. In Mediterranean areas (Portugal and Spain), several studies have shown that the interaction between domestic (in this case outdoor pigs) and wild animals leads to direct/indirect contact, which increases the risk of pathogens circulating in this interface, such as SE [19,20]. Other risk factors, such as the endemicity of the disease in the study area, seasonality and management practices with both domestic and wild animals, are key issues to take into account when assessing the risk of transmission of SE inter-species [20].

As was previously referred to, SE infection in wild boar may constitute a public health hazard due to its zoonotic potential [14].

In humans, infection with *Erysipelothrix rhusiopathiae* is mostly work-related (occupational exposure), including abattoir workers and veterinarians [3]. Erysipelas is the most common form of infection in humans [13], but a septic form with endocarditis or the generalised cutaneous form can also be observed [14]. Nowadays, this disease is considered to have an important zoonotic potential according to the scientific community, generating some concern in this regard [7].

Human infection occurs mostly through exposure to infected animals, their products (carcasses/viscera) or waste, or contaminated soil [13]. The entry route can include the ingestion of contaminated food or water and, particularly, small abrasions on the skin. Regarding SE in wild boar, the main risk group is the hunters that can be infected during evisceration and carcass preparation without using adequate protective devices.

For this reason, it is important to train hunters to adopt good hygiene and safety practices during the evisceration and preparation of wild game. All of these topics are included in the ‘Training of hunters in health and hygiene’ to undertake an initial examination of wild game on the spot, based on Regulation (EC) No 853/2004 [21].

This study aims to perform the first serological survey of *Erysipelothrix rhusiopathiae* infection in wild boars hunted for private consumption in Portugal.

## 2. Materials and Methods

### 2.1. Area of Study

This study took place in the district of Guarda, a traditional province of the “Beira-Alta” area in the Centre of Portugal, which represents 6% of the area of Portugal with an area of 551,434 hectares. Its territory is mostly mountainous with altitudes ranging from 84 and 1993 m (the highest point in mainland Portugal) and has bordering districts to the north, the district of Bragança, to east Spain, to the south district of Castelo-Branco and the districts of Viseu and Coimbra to the west.

The district of Guarda is a Portuguese Administrative region that integrates 14 Council and 243 Parishes mainly dedicated to agrarian production. In the district of Guarda, there are three different zones with distinct geographic and climatic features: sub-Atlantic, Atlantic-Mediterranean and sub-Mediterranean. These zones are beneficial for agricultural and animal production activities. The land cover is as follows: agriculture (34%), forested land (32%), shrub area (31), human (2.5%), and water (0.5%), as presented in Figure 1 [22].

The forested area provides an excellent environment for wildlife to flourish, leading to many hunting areas in the district, as shown in Figure 2. All hunting areas are unfenced and intensive hunting management with supplementary feeding feed for wild boar is not practiced.

The most frequent wild species in this district is the wild boar (*Sus scrofa*), which is an animal very resilient, opportunistic and moves very easily in search of food. Refuge areas such as forests are places where a large number of wild boar can be found, due to the optimum survival and staging characteristics that this species finds in these areas. However, they are omnivorous when it comes to food and seek out areas of cereals and crops [24].

As already mentioned, SE can circulate at the interface between domestic and wild pigs, with wild boar being a potential spillover or a spillback of the disease. For this reason, this study assessed the geographical distribution of pig farms in the study region with a view to identifying potential interface areas that could warrant greater attention and acuity [19,20]. The data depicted in Figure 3 were downloaded from the Portuguese National Statistical Institute and are related to official pig production [25]. Animal husbandry includes pigs’ farms (Figure 3), both in intensive and extensive production (outdoors), which can result in direct contact between pigs and wildlife, mainly wild boars. However, according to the Portuguese traditional rural way of life, many families use to have a pig in the backyard for self-consumption. This way, the real number of pigs must be higher than that presented in Figure 3.

### 2.2. Sampling and Laboratory Analysis

A non-probabilistic sampling method (convenience sampling) was used in this study. During the 2019/2020 hunting season (between October and February), hunting associations distributed throughout the district of Guarda were contacted to collaborate in the study. In total, 8 hunting associations accepted to participate. Written informed consent to take part in this scientific research was obtained from each association.

The samples were taken from wild boar hunted in organised hunts in 8 hunting areas in 4 counties in Guarda: Almeida (5), Guarda (1), Figueira de Castelo Rodrigo (1) and Pinhel (1). All samples were collected from wild boars legally hunted. No live animals were used for this study. This study did not involve the deliberate killing of animals. No ethical approval was deemed necessary.

In the study area, initial examination of hunted wild boar is not mandatory, and consequently, carcasses were not eviscerated on-spot.

Blood samples were obtained using a 10-mL syringe, tubes containing clot activator (BD Vacutainer^®^, Plymouth, UK) and an 80-mm long needle (1 × 280 mm, BOVIVET, Kruuse^®^, Langeskov, Denmark), using the Vacutainer System^®^ from the endocranial venous sinuses of the wild boars, as described by Arenas-Montes et al. (2013) [26]. Samples were refrigerated and transported in a cooling box to the laboratory. A total of 111 blood samples were collected. Each sample has associated data: location hunted (4 different counties: Pinhel, Figueira de Castelo Rodrigo, Almeida and Guarda), gender (42 males vs. 69 females) and age (102 adult vs. 9 subadult).

In the laboratory (2–3 h after collection), the blood samples were centrifugated (3500 r.p.m., 5 minutes) and serum stored at −20 °C until analyses.

The samples were analysed for antibodies against *Erysipelothrix rhusiopathiae* using a commercial kit (INgezin Mal Rojo^®^), an indirect monoclonal antibody ELISA technique specific for porcine immunoglobulins, following the manufacturer’s instructions. Briefly, this technique allows to distinguish serum samples as seropositive or seronegative on the basis of a cut off values that are established taking into account the optical density of positive and negative control supported in the ELISA kit.

### 2.3. Statistical Data Analysis

All data collected were gathered in a database using the Microsoft Excel^®^ program (version 16.0 Office 365), and the subsequent statistical analysis was performed using the JMP^®^ program (Student’s free license, 2021) and EpiTools^®^ (Ausvet, 2023). The absolute frequencies of disease categories in the study sample were calculated. The individual variation of seropositivity to *Erysipelothrix rhusiopathiae* obtained from hunted animals was evaluated to analyse the probability of this being related to age group, gender and origin. Comparisons of significant variables, assessed using the Chi-square test and Fisher’s Exact Test, were further explored using odds ratio (OR) estimates. The *p*-value was analysed for a confidence level higher than 95% (*p*-value < 0.05) [27].

## 3. Results

During the 2019/2020 hunting season, 111 wild boar blood samples were collected from 64 adult females, 38 adult males, 5 young females and 4 young males.

The Almeida County had the highest number of samples (80), followed by the Pinhel County with a total of 20 samples. In Figueira de Castelo Rodrigo County, eight samples were taken and, in Guarda County, three samples were collected.

Of the 111 wild boar serum samples tested for *Erysipelothrix rhusiopathiae*, 18 animals were positive, corresponding to an overall seroprevalence of 16.2% (95% CI: 19.9–24.4%). The 18 positive animals correspond to 2 young females (1.8% (95% CI: 0.2–6.4%)), 7 adult males (6.3% (95% CI: 2.6–12.6%)) and 9 adult females (8.1% (95% CI: 3.8–14.8%)). When analysing the statistical relationship between the gender of the animals and the seropositivity of the results, as well as the relationship between the age of the analysed animals and the same laboratory results, no significant differences were found (confidence interval of 95%; *p*-value < 0.05).

In this study, different seroprevalence were observed in the geographical areas evaluated. Wild boar hunted in Pinhel (*n* = 20) presented the highest seroprevalence (40%, 95% CI: 19.1–64%), followed by Guarda (*n* = 3), which presented a seroprevalence of 33.3% (95% CI: 0.8–90.6%). In Figueira Castelo Rodrigo County, wild boar (*n* = 8) presented and seroprevalence of 12.5% (95% CI: 0.3–52.7%), and in Almeida County (*n* = 80) it was observed an overall seroprevalence of 10% (95% CI: 4.4–18.8%) (Figure 4).

When statistically analysed, Pinhel was the only county where it is possible to observe significant statistical differences (*p*-value < 0.05), with an animal seropositive for *Erysipelothrix rhusiopathiae* being five times more likely to be hunted in this county (Odds ratio = 5.4).

Due to the large number of farmers who raise pigs, either as a complementary agricultural activity (see Figure 3) or as a form of self-consumption, the diseases of wild animals, particularly wild boar, can easily be transmitted to domestic livestock. The most recent agrarian census [25] showed a pigs’ production ranging between 1 and 500 animals per Parish (Figure 3). When analysing the overlapping between seroprevalence of *Erysipelothrix rhusiopathiae* in wild boar and the number of pigs per farm, it is possible to see that the Counties of Guarda, Pinhel, Almeida and Figueira de Castelo Rodrigo revealed the existence of potential interfaces for the transmission of the disease to domestic pigs.

## 4. Discussion

Hunting is an activity of major social, cultural and economic importance in Portugal, with wild boar being the most hunted game species [28]. Knowledge of the existence of diseases shared between wild game animals, livestock and humans is extremely important for taking effective measures to control, mitigate and manage them effectively.

In this study, an *Erysipelothrix rhusiophatie* seroprevalence of 16.2% (95% CI: 19.9–24.4%) was observed, highlighting the importance of this zoonotic shared infection in the wild boar population. Since this was the first time that the seroprevalence of *Erysipelothrix rhusiophatie* in wild boars was carried out in Portugal, there is no possibility of making comparisons. As in Europe, in the rest of the world, information on infection in wild populations, especially wild boars, is practically nil. It is possible, however, to compare with serological tests carried out in Spain several years ago, in which two surveys on viral and bacterial pathogens indicate that the seroprevalence to *Erysipelothrix rhusiophatie* ranges between 5% in 2002 [29] and 15% in 2012 [30] on Spanish wild boars. In another extended Spanish long-term study, the *Erysipelothrix rhusiophatie* seropositivity was monitored, and in 7 of the 10 years it was present, considering this infection as endemic, scarcely active and spreading [31]. Outside Europe, only in Japan [32], some studies refer to wild boars as potential sources of erysipelas infection, with 95.6% de seroprevalence in the tested animals [30] and with confirmed, antibiotic resistance [33,34].

The presence of wild boar with SE can represent a risk of transmission to sympatric populations of domestic pigs, especially on extensive farms where natural resources are shared between domestic and wild populations. The geographical distribution of pig farms in the study region revealed potential interface areas between pig farms (Figure 3), both in intensive and extensive (outdoor) production, which could result in direct or indirect contact with seropositive wild boar in the municipalities of Almeida, Guarda, Figueira de Castelo Rodrigo and Pinhel.

Although in this study we did not assess risk factors for disease transmission between domestic and wild populations, it is known that wild boar is an animal that is very close to human areas, especially in the aforementioned cultivated areas, areas where animals are reared and where there are ponds [35,36]. It is suggested that future studies be carried out to assess the transmission potential of this disease, especially in Pinhel County, where a higher seroprevalence was observed in wild boar. However, wild boar, due to their ability to colonise any habitat and to move around, can play a special role in the dispersal of diseases [32,36], resulting in contact between domestic and other wild animals in geographical areas bordering those assessed in this study. As mentioned above, the sampling was of convenience and was limited to evaluating wild boar hunted in a small geographical area, so it is suggested that this study be extended to other geographical areas, particularly where there is evidence of an interface between wild boar and domestic pigs.

Furthermore, since *Erysipelothrix rhusiophatie* may affect many species of wild and domestic mammals and birds as well as reptiles, amphibians and fish [12], additional studies could be carried out to assess the existence of this disease in other animals and analyse the level of its spread. To date, in Portugal, according to the authors’ knowledge, there are no scientific records of the presence of this agent in other animals. With this information, seroprevalence results of a zoonotic infectious disease above 15% are alarming, mainly because the sampling was carried out in an area of Portugal where hunted animals are not obligatorily subject to on-spot initial examination when meat is used for private consumption [37]. Since carcasses are not eviscerated and submitted to the on-spot initial examination, there is a gap in the information regarding the health status of these animals. The correct initial examination of wild boar [37,38] could allow the identification of lesions compatible with *Erysipelothrix rhusiophatiae* infection on the carcasses of hunted wild boar. Hunters could thus be alerted and effective control measures and means of preventing the spread of the infection in man and animals could be adopted in a timely manner.

Furthermore, all hunters transported the non-eviscerated animals to their homes for private consumption. Subsequently, wild boars are eviscerated, sometimes without proper hygiene and biosecurity precautions, increasing the potential risk of exposure [13]. Since human infection occurs mainly through skin abrasions, the use of protective gloves is of utmost importance to mitigate the risk of exposure.

Furthermore, by-products may be disposed of, sometimes not in an appropriate manner. As emphasised by Vieira-Pinto et al. (2011), the proper disposal of by-products is a capital measure to mitigate the spread of various diseases, which can affect not only game animals, other wild animals (including protected or endangered ones such as the lynx and the Iberian wolf), but also domestic animals and humans [37]. Since *Erysipelothrix rhusiophatie* is resistant and has the ability to grow in dead animal tissues and decaying organic matter [14], this bad practice may favour the disease spread, as also described by Abrantes et al. (2023) for the case of tuberculosis [39].

This infection, as a zoonosis, is mainly an occupational disease, thus posing a health risk to those who come into direct contact with infected animals [14,40,41]. Therefore, in the context of SE in wild boar, informing and training hunters (the main risk group) is of key importance for their protection and to minimize disease spread. Under this context, the ‘Training of hunters in health and hygiene’ defined by Regulation 853/2004, seems to represent a capital tool [21].

## 5. Conclusions

A seroprevalence of 16.2% (95% CI: 19.9–24.4%) was observed in this study. The impact of diseases such as erysipelas can compromise the wild boar population, mainly due to its chronic and debilitating nature. In addition, SE in wild boar may pose a potential risk of zoonotic exposure, especially for hunters, when handling wild boar carcasses without hygienic and biosecurity care.

Training for hunters is necessary to raise awareness to the risk of occupational exposure associated with this and other zoonotic diseases. Under this concept, the veterinarian role has significant relevance.

The results raise awareness among actors at the interface between wildlife, livestock and humans (One Health approach) on the increasing importance of implementing synergistic efforts on health management/control of shared infection diseases, such as SE. Furthermore, the surveillance plan for large game diseases should also be strengthened to monitor the evolution of this disease in the wild boar population aiming to mitigate the negative impact of this disease on human and animal health.

## Figures and Tables

**Figure 1 animals-13-02936-f001:**
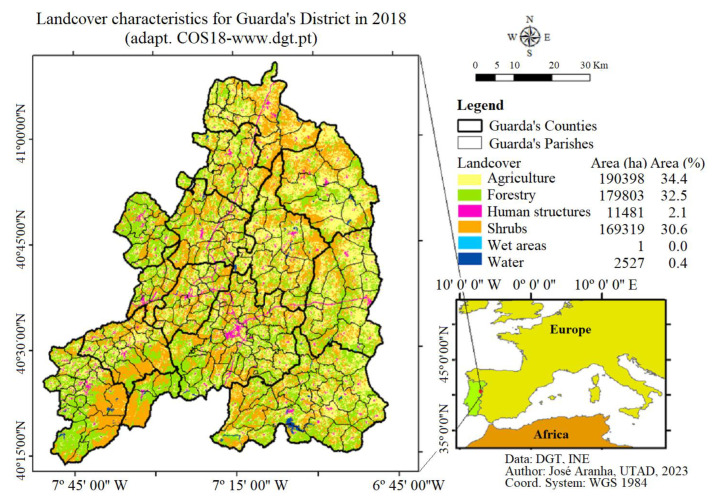
Landcover related to the district of Guarda in 2018 (adapted from [22] COS18: www.dgt.pt).

**Figure 2 animals-13-02936-f002:**
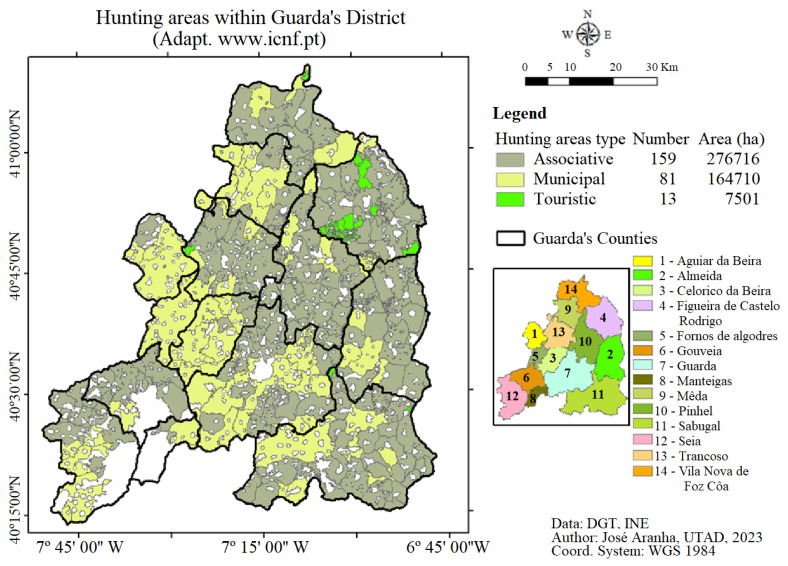
Hunting areas that exist in the district of Guarda (adapted from [23] www.icnf.pt).

**Figure 3 animals-13-02936-f003:**
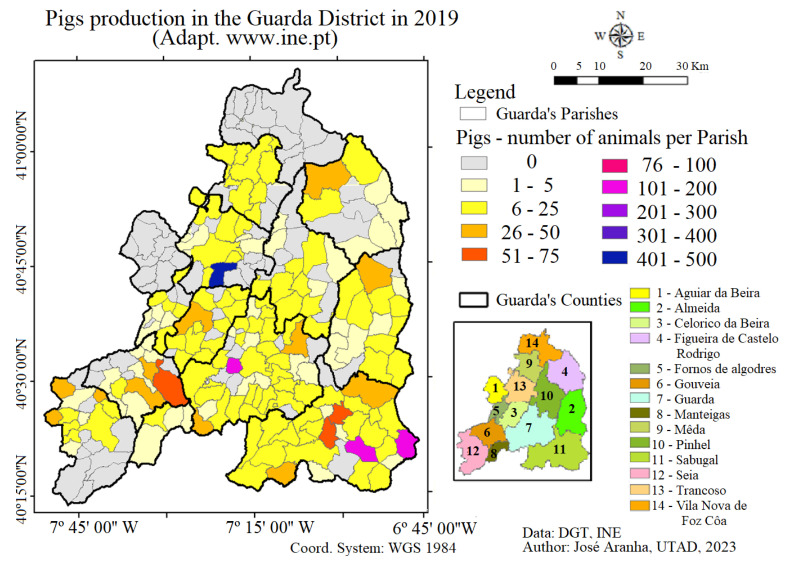
Pigs production in the Guarda District (adapted from [25] www.ine.pt).

**Figure 4 animals-13-02936-f004:**
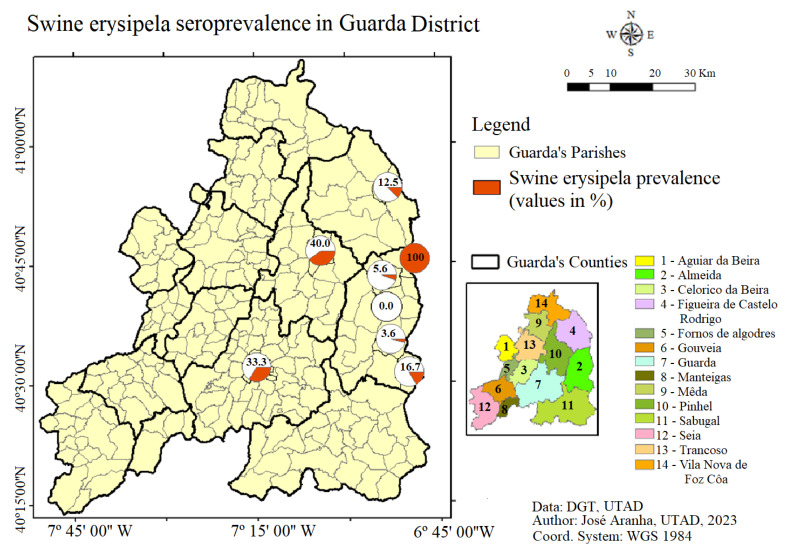
Schematic results of the seroprevalence of *Erysipelothrix rhusiopathiae* in the wild boars analysed in the Guarda district.

## Data Availability

Data sharing not applicable.

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
