# Peer review of "First Serologic Survey of Erysipelothrix rhusiopathiae in Wild Boars Hunted for Private Consumption in Portugal"

_animals, 2023, doi:10.3390/ani13182936_

Round 1

Reviewer 1 Report

The work carried out provides the first epidemiological study on an important zoonotic agent such as Erysipelothrix rhusiopathiae in a small portion of Portugal. knowledge in veterinary and human medicine is extremely important, in order to know the risk of infection.

The development and methodology used are adequate. However, numerous errors have been detected in the document.

Line 23: "ES" must be corrected.

Line 81: The abbreviation "E. rhusiopathiae" is used, but later in the rest of the text it is not used again. It is recommended to unify the criteria.

The figure located in the document does not correspond to the one attached. In addition, the denominations are not correctly understood. The exposed numerical results do not correspond to those shown in the text. It is recommended to correct and better explain said figure.

In my opinion, lines 192-200 should be included in the discussion section and not in the conclusion. In addition, the seropositivity result obtained in the conclusion section should be indicated.

Likewise, a section on the main limitations of the study should be added since the research that has been carried out only represents 6% of the surface of the Portuguese territory. In addition, the small sample size in this territory is not uniform or homogeneous. The background of the animals used is not known. The specificity and sensitivity of the test used is unknown and it has not been compared with other diagnostic techniques, etc.

References 2, 7, 12 and 13 must be reviewed and corrected to unify the exposed bibliography.

Author Response

Dear Reviewer,

The authors would like to thanks this opportunity for improving the paper according to the valuable comments given.

We have uploaded a version of our manuscript, upgraded to an article, with all changes tracked, according to reviewers’ comments. Please see the attachment.

In the following text, the authors respond to the comments made by the reviewer, being as specific as possible.

We hope that this work may be considered suitable for publication in the ANIMALS journal.

Thank you in advance for the time and effort that will be involved in reviewing this work. 

Yours sincerely,

Madalena Vieira-Pinto

Reviewer's comments: The work carried out provides the first epidemiological study on an important zoonotic agent such as Erysipelothrix rhusiopathiae in a small portion of Portugal. knowledge in veterinary and human medicine is extremely important, in order to know the risk of infection.

Answer: The authors are grateful for the positive comment on their study.

Reviewer's comments: The development and methodology used are adequate. However, numerous errors have been detected in the document. Line 23: "ES" must be corrected.

Answer: ES was revised according to reviewer suggestion.

Reviewer's comments: Line 81: The abbreviation "E. rhusiopathiae" is used, but later in the rest of the text it is not used again. It is recommended to unify the criteria.

Answer: Erysipelothrix rhusiopathiae has been harmonised throughout the text.

Reviewer's comments: The figure located in the document does not correspond to the one attached. In addition, the denominations are not correctly understood. The exposed numerical results do not correspond to those shown in the text. It is recommended to correct and better explain said figure.

Answer: Figure and text was revised according to reviewer suggestion.

Reviewer's comments: In my opinion, lines 192-200 should be included in the discussion section and not in the conclusion. In addition, the seropositivity result obtained in the conclusion section should be indicated.

Answer: Text was changed according to reviewer suggestion.

Reviewer's comments: Likewise, a section on the main limitations of the study should be added since the research that has been carried out only represents 6% of the surface of the Portuguese territory. In addition, the small sample size in this territory is not uniform or homogeneous. The background of the animals used is not known. The specificity and sensitivity of the test used is unknown and it has not been compared with other diagnostic techniques, etc.

Answer: Some limitations were added to the text.

Reviewer's comments: References 2, 7, 12 and 13 must be reviewed and corrected to unify the exposed bibliography.

Answer: All the references have been updated and renumbered since new ones were included in this revised version.

Reviewer 2 Report

The article presents novel interesting information about a pathogen of animal and public health importance and interest. However, quality of the presentation of data, as well as the depth of the data provided in the introduction and discussion is  poor.

General comments:

Introduction

A deeper review about about occurrence of SE in wild boar as well as other wild species from other countries mostly in Europe, as well as occurrence of SE in domestic species in Portugal is necessary.

More information about the epidemiology of the pathogen is necessary, e.g. how the chronic animals maintain the diseases?

More information about wild boar ecology is necessary (e.g. Wild boar home range, wild boar population density and other population ecology information of this species in Portugal. Which kind of wild boar population managements use to be used in Portugal?

Material and Methods 

Study area:

How many hectares were sampled? Which kind of land cover is represented in the study area? Which kind of climate is representative of this area?

How many domestic pigs and other domestic SE susceptible species inhabit in the study area?

Which other wild susceptible species are present in the study area?

Are those hunting areas fenced? Some studies have found lower prevalence in fenced areas comparing with open areas. You should specify this information?

There exists supplementary feeding for wild boar in the study area?

How is the estimated density of wild boar population in the study area?

Why did you select this area?

Sampling and laboratory analysis:

Information about specific dates (months) of sampling season in each year is necessary.

Information about number of males and females, adults and juveniles sampled is necessary in this section.

Specific information (time and g) about the centrifugation process is necessary. How long does it take from sampling time until it is centrifuged?

How did you store the serum samples until its analysis?

More information about ELISA technique is necessary. Which mean antibody titre in the ELISA did you used as threshold? 

Statistical analysis

You should specify the number of age classes and origin clases that you are analyzing. Take into account that you cannot use student T-test for the analysis of variables composed of more than two categories.

Did you analyzed the influence of the year or hunting season in the results? 

Results.

You should add the value of the statistical test employed as well as the confidence interval obtained. 

Discussion.

In general, the discussion is too brief. Even if is is the first time the pathogen is detected in wild boars in Portugal, more information about its prevalence in other wild and domestic species in Portugal and Europe is necessary.

You mentioned that hunting is an important social, cultural and economic activity in Portugal but there are no data or references supporting this sentence.

You obtained that Pinhel county presented the higher seroprevalence. Which could be the explanation of this results? You should add this information in the discussion

Some editions about english language are required.

Author Response

Dear Reviewer,

The authors would like to thanks this opportunity for improving the paper according to the valuable comments given.

We have uploaded a version of our manuscript, upgraded to an article, with all changes tracked, according to reviewers’ comments. Please see the attachment.

In the following text, the authors respond to the comments made by the reviewer, being as specific as possible.

We hope that this work may be considered suitable for publication in the ANIMALS journal.

Thank you in advance for the time and effort that will be involved in reviewing this work. 

Yours sincerely,

Madalena Vieira-Pinto

Reviewer's comments: The article presents novel interesting information about a pathogen of animal and public health importance and interest. However, quality of the presentation of data, as well as the depth of the data provided in the introduction and discussion is  poor.

Answer: The authors are grateful for the comment on their study.

Reviewer's comments: More information about the epidemiology of the pathogen is necessary, e.g. how the chronic animals maintain the diseases?

Answer: Two sentences was added about this topic in the Introduction Section.

Reviewer's comments: More information about wild boar ecology is necessary (e.g. Wild boar home range, wild boar population density and other population ecology information of this species in Portugal. Which kind of wild boar population managements use to be used in Portugal?

Answer: This information was added to several sections of the manuscript. Density and ecology information in the introduction and the type of management in the M&M, for example.

General comments - Material and Methods - Study area - Reviewer's comments: How many hectares were sampled? Which kind of land cover is represented in the study area? Which kind

Answer: Additional data about the study area was included in Material and Methods.

Material and Methods - Study area - Reviewer's comments: How many domestic pigs and other domestic SE susceptible species inhabit in the study area?

Answer: Figure 3 was added to the text presenting the geographical distribution of pig farms in the study region with a view to identifying potential interface areas that could warrant greater attention and acuity. Since SE may affect different other species (mammals and birds) it is not possible to present data about the existence of all other domestic SE susceptible species inhabit in the study area, as suggested by the reviewer. Since pigs are considered the main reservoir of SE, the authors would like to focus the article on these two species: wild boar and domestic pigs. We would very grateful if the reviewer could understand our perspective.

Reviewer's comments: Which other wild susceptible species are present in the study area?

Answer: As referred in the sentence above, E. rhusiopathiae may affect many species of wild and domestic mammals and birds as well as reptiles, amphibians, and fish…It is impossible to collect all this information to add to this article. For that reason, the authors would like to focus on the main reservoir of this infectious disease: wild boar and domestic pigs.

Reviewer's comments: Are those hunting areas fenced? Some studies have found lower prevalence in fenced areas comparing with open areas. You should specify this information?

Answer: Information was specified in M&M Section: “All hunting areas are unfenced and intensive hunting management with  supplementary feeding feed for wild boar is not practiced.”

Reviewer's comments:There exists supplementary feeding for wild boar in the study area?

Answer: A new sentence was added to M& M Section: “All hunting areas are unfenced and intensive hunting management with  supplementary feeding feed for wild boar is not practiced.”

Reviewer's comments: How is the estimated density of wild boar population in the study area?

Answer: There no data on the density of wild boar population in the study area. For that reason, it was impossible to include it in this study.

Reviewer's comments: Why did you select this area?

Answer: It is an important hunting area in Portugal and the first author of this study lives in Guarda, having access to hunting associations to ask for their participation in this study and to allow the sample collection during the driven hunts.

Reviewer's comments: Sampling and laboratory analysis - Information about specific dates (months) of sampling season in each year is necessary.

Answer: The authors thanks for the suggestion and update the information in the section M&M part area of study.

Reviewer's comments: Sampling and laboratory analysis Information about number of males and females, adults and juveniles sampled is necessary in this section.

Answer: This information was added in the M&M section

Reviewer's comments - Sampling and laboratory analysis: Specific information (time and g) about the centrifugation process is necessary. How long does it take from sampling time until it is centrifuged?

Answer: This information was added in the M&M section.

Reviewer's comments - Sampling and laboratory analysis: How did you store the serum samples until its analysis?

Answer: This information was added in the M&M section.

Reviewer's comments - Sampling and laboratory analysis: More information about ELISA technique is necessary. Which mean antibody titre in the ELISA did you used as threshold?

Answer: The following sentence was added: “In the laboratory, after centrifugation (3500 r.p.m., five minutes), the serum samples were analysed for antibodies against Erysipelothrix rhusiopathiae using a commercial kit (INgezin Mal Rojo®), an indirect monoclonal antibody ELISA technique specific for porcine immunoglobulins, following the manufacturer's instructions. Briefly, this technique allows to distinguish serum samples as seropositive or seronegative on the basis of a cut off values that are established taking into account the optical density of positive and negative control supported in the ELISA kit”.

Reviewer's comments-  Statistical analysis - You should specify the number of age classes and origin classes that you are analyzing. Take into account that you cannot use student T-test for the analysis of variables composed of more than two categories. Did you analyzed the influence of the year or hunting season in the results?

Answer: The authors apologise for the error in the presentation of the statistics.  In order to compare all the municipalities under study together, the Chi-square test was used. And Fischer's exact test in specific cases in order to compare the municipalities under study individually, guaranteeing correct OD values. The error was corrected in the paper.

The remain information was added in the M&M section. No analyses regarding influence of the year or hunting season were done, because sampling was performed only during three months (between October and February) throughout the 2019/2020 hunting season, as it is referred in M&M section

Reviewer's commentsResults - You should add the value of the statistical test employed as well as the confidence interval obtained.

Answer: This information was added in the results section

Reviewer's comments - You mentioned that hunting is an important social, cultural and economic activity in Portugal but there are no data or references supporting this sentence.

Answer: This sentence was moved to the discussion as recommended by reviewer 1. Reference was included

Reviewer's comments - You obtained that Pinhel county presented the higher seroprevalence. Which could be the explanation of this results? You should add this information in the discussion

Answer: The authors do not have a answer for this important question. For this reason it was added two paragraphs in the discussion underlining 
